# DEEP GATED CANONICAL CORRELATION ANALYSIS

## ABSTRACT

Canonical Correlation Analysis (CCA) models can extract informative correlated representations from multimodal unlabelled data. Despite their success, CCA models may break if the number of variables exceeds the number of samples. We propose Deep Gated-CCA, a method for learning correlated representations based on a sparse subset of variables from two observed modalities. The proposed procedure learns two non-linear transformations and simultaneously gates the input variables to identify a subset of most correlated variables. The non-linear transformations are learned by training two neural networks to maximize a shared correlation loss defined based on their outputs. Gating is obtained by adding an approximate $\ell_0$ regularization term applied to the input variables. This approximation relies on a recently proposed continuous Gaussian based relaxation for Bernoulli variables which act as gates. We demonstrate the efficacy of the method using several synthetic and real examples. Most notably, the method outperforms other linear and non-linear CCA models.

## 1 INTRODUCTION

Canonical Correlation Analysis (CCA) (Hotelling, 1936; Thompson, 2005), is a classic statistical method for finding the maximally correlated linear transformations of two modalities (or views). Using modalities $\boldsymbol{X} \in \mathbb{R}^{D_x \times N}$ and $\boldsymbol{Y} \in \mathbb{R}^{D_Y \times N}$, which are centered and have $N$ samples with $D_x$ and $D_y$ features respectively. CCA seeks canonical vectors $\boldsymbol{a_i} \in \mathbb{R}^{D_X}$, and $\boldsymbol{b_i} \in \mathbb{R}^{D_Y}$, such that, $\boldsymbol{u_i} = \boldsymbol{a_i^T X}$, and $\boldsymbol{v_i} = \boldsymbol{b_i^T Y}$, $i = 1, ..., N$, maximize the sample correlations between $\boldsymbol{u_i}$ and $\boldsymbol{v_i}$, where $\boldsymbol{u_i}$ ($\boldsymbol{v_i}$) form an orthonormal basis for $i = 1, ..., d$, i.e.

$$\boldsymbol{a_i}, \boldsymbol{b_i} = \underset{\langle \boldsymbol{u_i}, \boldsymbol{u_j} \rangle = \delta_{i,j}, \langle \boldsymbol{v_i}, \boldsymbol{v_j} \rangle = \delta_{i,j}, i,j=1,...,d}{\operatorname{argmax}} \operatorname{Corr}(\boldsymbol{u_i}, \boldsymbol{v_i}). \tag{1}$$

While CCA enjoys a closed-form solution using a generalized eigen pair problem, it is restricted to the linear transformations $\boldsymbol{A} = [\boldsymbol{a}_1, ..., \boldsymbol{a}_d]$ and $\boldsymbol{B} = [\boldsymbol{b}_1, ..., \boldsymbol{b}_d]$.

In order to identify non-linear relations between input variables, several extensions of CCA have been proposed. Kernel methods such as Kernel CCA (Bach & Jordan, 2002), Non-paramatric CCA (Michaeli et al., 2016) or Multi-view Diffusion maps (Lindenbaum et al., 2020) learn the non-linear relations in reproducing Hilbert spaces. These methods have several shortcomings: they are limited to a designed kernel, they require $\mathcal{O}(N^2)$ computations for training, and they have poor interpolation and extrapolation capabilities. To overcome these limitations, Andrew et al. (2013) have proposed Deep CCA, to learn parametric non-linear transformations of the input modalities $\boldsymbol{X}$ and $\boldsymbol{Y}$. The functions are learned by training two neural networks to maximize the total correlation between their outputs.

Linear and non-linear canonical correlation models have been widely used in the setting of unsupervised or semi-supervised learning. When $d$ is set to a dimension satisfying $d < D_x, D_y$, these models find dimensional reduced representations that may be useful for clustering, classification or manifold learning in many applications. For example, in biology (Pimentel et al., 2018), neuroscience (Al-Shargie et al., 2017), medicine (Zhang et al., 2017), and engineering (Chen et al., 2017). One key limitations of these models is that they typically require more samples than features, i.e. $N > D_x, D_y$. However, if we have more variable than samples, the estimation based on the closed form solution of the CCA problem (in Eq. 1) breaks (Suo et al., 2017). Moreover, in high dimensional data, often some of the variables are not informative and thus should be omitted

from the transformations. For these reasons, there has been a growing interest in studying sparse CCA models.

Sparse CCA (SCCA) (Waaijenborg et al., 2008; Hardoon & Shawe-Taylor, 2011; Suo et al., 2017) uses an $\ell_1$ penalty to encourage sparsity of the canonical vectors $a_i$ and $b_i$. This can not only remove the degeneracy inherit to $N < D_x, D_y$, but can improve interpetability and performance. One caveat of this approach is its high computational complexity, which can be reduced by replacing the orthonormality constraints on $u_i$ and $v_i$ with orthonormality constraints on $a_i$ and $b_i$. This procedure is known as simplified-SCCA (Parkhomenko et al., 2009; Witten et al., 2009), which enjoys a closed form solution. There has been limited work on extending these models to sparse non-linear CCA. Specifically, there are two kernel based extensions, two-stage kernel CCA (TSKCCA) by Yoshida et al. (2017) and SCCA based on Hilbert-Schmidt Independence Criterion (SCCA-HSIC) by Uurtio et al. (2018). However, these models suffer from the same limitations as KCCA and are not scalable to a high dimensional regime.

This paper presents a sparse CCA model that can be optimized using standard deep learning methodologies. The method combines the differentiable loss presented in DCCA (Andrew et al., 2013) with an approximate $\ell_0$ regularization term designed to sparsify the input variables of both $X$ and $Y$. Our regularization relies on a recently proposed Gaussian based continuous relaxation of Bernoulli random variables, termed gates (Yamada et al., 2020). The gates are applied to the input features to sparsify $X$ and $Y$. The gates parameters are trained jointly via stochastic gradient decent to maximize the correlation between the representations of $X$ and $Y$, while simultaneously selecting only the subsets of the most correlated input features. We apply the proposed method to synthetic data, and demonstrate that our method can improve the estimation of the canonical vectors compared with SCCA models. Then, we use the method to identify informative variable in multi-channel noisy seismic data and show its advantage over other CCA models.

## 1.1 BACKGROUND

## 1.2 DEEP CCA

Andrew et al. (2013), present a deep neural network that learns correlated representations. They proposed Deep Canonical Correlation Analysis (DCCA) which extracts two nonlinear transformations of $X$ and $Y$ with maximal correlation. DCCA trains two neural networks with a joint loss aiming to maximize the total correlation of the network's outputs. The parameters of the networks are learned by applying stochastic gradient decent to the following objective:

$$\theta_X^*, \theta_Y^* = \underset{\theta_X, \theta_Y}{\operatorname{argmax}} \operatorname{Corr}(f(X; \theta_X), g(Y; \theta_Y)), \tag{2}$$

where $\theta_X$ and $\theta_Y$ are the trainable parameters, and $f(X), g(Y) \in \mathbb{R}^d$ are the desired correlated representations.

## 1.3 SPARSE CCA

Several authors have proposed solutions for the problem of recovering sparse canonical vectors. The key advantages of sparse vectors are that they enable identifying correlated representations even in the regime of $N < D_x, D_y$ and they allow unsupervised feature selection. Following the formulation by Suo et al. (2017), SCCA could be described using the following regularized objective

$$a, b = \operatorname{argmin} \left[ -\operatorname{Cov}(a^T X, b^T Y) + \tau_1 \|a\|_1 + \tau_2 \|b\|_1 \right],$$
$$\text{subject to} \quad \|a^T X\|_2 \leq 1, \quad \|b^T Y\|_2 \leq 1,$$

where $\tau_1$ and $\tau_2$ are regularization parameters for controlling the sparsity of the canonical vectors $a$ and $b$. Note that the relaxed inequality constrain on $a^T X$ and $b^T Y$ makes the problem bi-convex, however, if $\|a^T X\|_2 < 1$ or $\|b^T X\|_2 < 1$, then the covariance in the objective is no longer equal to the correlation.

## 1.4 STOCHASTIC GATES

In the last few years, several methods have been proposed for incorporating discrete random variables into gradient based optimization methods. Towards this goal, continuous relaxations of discrete random variables such as (Maddison et al., 2016; Jang et al., 2017) have been proposed.

Such relaxations have been used in several applications, for example, model compression (Louizos et al., 2017), feature selection or for defining discrete activations (Jang et al., 2016). We focus on a Gaussian-based relaxation of Bernoulli variables, termed Stochastic Gates (STG) (Yamada et al., 2020), which were originally proposed for supervised feature selection. We denote the STG random vector by $\mathbf{z} \in [0, 1]^D$, where each entry is defined as

$$\mathbf{z}[i] = \max(0, \min(1, \mu[i] + \epsilon[i])), \tag{3}$$

where $\mu[i]$ is a trainable parameter for entry $i$, the injected noise $\epsilon[i]$ is drawn from $\mathcal{N}(0, \sigma^2)$ and $\sigma$ is fixed throughout training. This approximation can be viewed as a clipped, mean-shifted, Gaussian random variable. In Fig. 1 we illustrate generation of the transformed random variable $\mathbf{z}[i]$ for $\mu[i] = 0.5$ which represents a "fair" relaxed Bernoulli variable.

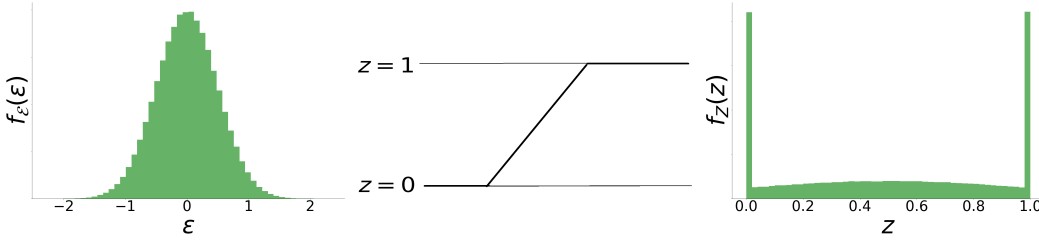

**Figure 1:** From left to right, pdf of the Gaussian injected noise $\epsilon$, the hard Sigmoid function (defined in Eq. 3) and the pdf of the relaxed Bernoulli variable for $\mu = 0.5$ corresponding to a "fair" Bernoulli variable. The trainable parameter $\mu$ can shift the mass of z towards 0 or 1. Here, we refer to one element of the random vector and omit the index $i$.

## 2 DEEP GATED CCA

### 2.1 MODEL

It is appealing to try to combine ideas from Sparse CCA into the rich differentiable model of Deep CCA. However, a straight forward $\ell_1$ regularization of the input layer of a neural network does not work in practice because it makes the learning procedure unstable. This was observed in the supervised setting by Li et al. (2016); Feng & Simon (2017). This instability occurs because the objective is not differentiable everywhere. To overcome this limitation, we use the STG random variables (see Eq. 3) by multiplying them with the features of $\boldsymbol{X}$ and $\boldsymbol{Y}$. Then, by penalizing for active gates using a regularization term $\mathbb{E}\|\mathbf{z}\|_0$, we can induce sparsity in the input variables.

We formulate the problem of sparse nonlinear CCA by regularizing a deep neural network with a correlation term. We introduce two random STG vectors into the input layers of two neural networks which are trained in tandem to maximize the total correlation. Denoting the random gating vectors $\mathbf{z}_x$ and $\mathbf{z}_y$ for view $\boldsymbol{X}$ and $\boldsymbol{Y}$ respectively, the Deep Gated CCA (DG-CCA) loss is defined by

$$L(\boldsymbol{\theta}, \boldsymbol{\mu}) = \mathbb{E}_{\mathbf{z}_x, \mathbf{z}_y} \left[ -\text{Corr}(\boldsymbol{f}(\mathbf{z}_x^T \boldsymbol{X}; \boldsymbol{\theta_X}), \boldsymbol{g}(\mathbf{z}_y^T \boldsymbol{Y}; \boldsymbol{\theta_Y})) + \lambda_x \|\mathbf{z}_x\|_0 + \lambda_y \|\mathbf{z}_y\|_0 \right], \tag{4}$$

where $\boldsymbol{\theta} = (\boldsymbol{\theta_X}, \boldsymbol{\theta_Y})$, $\boldsymbol{\mu} = (\boldsymbol{\mu_X}, \boldsymbol{\mu_Y})$ are the model parameters, and $\lambda_x$, $\lambda_y$ are regularization parameters that control the sparsity of the input variables. The vectors $\mathbf{z}_x$ and $\mathbf{z}_y$ are random STG vectors, with elements defined based on Eq. 3.

Fig. 2 highlights the proposed architecture. Each observed modality is first passed through the gates. Then, the outputs of the gates are used as inputs to a view-specific neural sub-net. Finally, the shared loss term in Eq. 4 is minimized by optimizing the parameters of the gates and neural sub-nets.

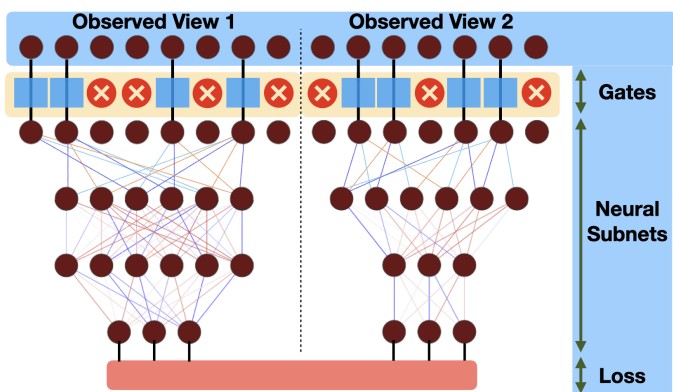

**Figure 2:** The proposed architecture. Data from two views is propagated through stochastic gates. The gates output is fed into two neural sub-nets that have a shared loss. The shared loss is computed on both neural sub-nets output representations (with dimension $d = 3$ in this example). The shared loss combines a total correlation term and a differentiable regularization term which induces sparsity in the input variables.

## 2.2 ALGORITHM DETAILS

We now detail the procedure used in DG-CCA for minimizing the loss $L(\boldsymbol{\theta}, \boldsymbol{\mu})$ (in Eq. 4). The regularization is based on a parametric expectation and therefore, can be expressed by

$$\mathbb{E}_{\mathbf{z}_x} \|\mathbf{z}\|_0 = \sum_{i=1}^{D_x} \mathbb{P}(\mathbf{z}_x[i] \geq 0) = \sum_{i=1}^{D_x} \left( \frac{1}{2} - \frac{1}{2} \operatorname{erf}\left( -\frac{\mu_x[i]}{\sqrt{2}\sigma} \right) \right),$$

where $\operatorname{erf}()$ is the Gaussian error function, and is defined similarly for $\mathbb{E}_{\mathbf{z}_y} \|\mathbf{z}_y\|_0$. Denoting the centered output representations of $\boldsymbol{X}, \boldsymbol{Y}$ by $\boldsymbol{\Psi}_X, \boldsymbol{\Psi}_Y \in \mathbb{R}^{d \times N}$ respectively, the empirical covariance matrix between these representations can be expressed as $\widehat{\boldsymbol{\Sigma}}_{XY} = \frac{1}{N-1} \boldsymbol{\Psi}_X \boldsymbol{\Psi}_Y^T$. Using a similar notations, we express the regularized empirical covariance matrices of $\boldsymbol{X}$ and $\boldsymbol{Y}$ as $\widehat{\boldsymbol{\Sigma}}_X = \frac{1}{N-1} \boldsymbol{\Psi}_X \boldsymbol{\Psi}_X^T + \gamma \boldsymbol{I}$ and $\widehat{\boldsymbol{\Sigma}}_Y = \frac{1}{N-1} \boldsymbol{\Psi}_Y \boldsymbol{\Psi}_Y^T + \gamma \boldsymbol{I}$, where the matrix $\gamma \boldsymbol{I}$ ($\gamma > 0$) is added to stabilize the invertability of $\widehat{\boldsymbol{\Sigma}}_X$ and $\widehat{\boldsymbol{\Sigma}}_Y$. The total correlation in Eq. 4 can be expressed using the trace of $\widehat{\boldsymbol{\Sigma}}_Y^{-1/2} \widehat{\boldsymbol{\Sigma}}_{YX} \widehat{\boldsymbol{\Sigma}}_X^{-1} \widehat{\boldsymbol{\Sigma}}_{XY} \widehat{\boldsymbol{\Sigma}}_Y^{-1/2}$.

To learn the parameters of the gates $\boldsymbol{\mu}$ and of the representations $\boldsymbol{\theta}$ we apply stochastic gradient decent to $L(\boldsymbol{\theta}, \boldsymbol{\mu})$. Specifically, we used Monte Carlo sampling to estimate the left part of Eq. 4. This is repeated for each batch, using one Monte Carlo sample per batch as suggested by Louizos et al. (2017) and Yamada et al. (2020), and worked well in our experiments. After training we remove the stochastic part of the gates, and use only variables $i_x \in \{1, ..., D_X\}$ and $i_y \in \{1, ..., D_y\}$ such that $\mathbf{z}_x[i_x] > 0$ and $\mathbf{z}_y[i_y] > 0$.

## 3 RESULTS

In the following section we detail the evaluation of the proposed approach using synthetic and real datasets. We start with two linear examples, demonstrating the performance of DG-CCA when $N \ll D_X, D_Y$. Then, we use noisy images from MNIST and seismic data measured using two channels to demonstrate that DG-CCA finds meaningful representations of data even in a noisy regime. For a full description of the training procedure as well as the baseline methods, we refer the reader to the Appendix.

### 3.1 SYNTHETIC EXAMPLE

We start by describing a simple linear model also experimented by Suo et al. (2017). Consider data generated from the following distribution $\begin{pmatrix} \boldsymbol{X} \\ \boldsymbol{Y} \end{pmatrix} \sim N(\begin{pmatrix} \boldsymbol{0} \\ \boldsymbol{0} \end{pmatrix}, \begin{pmatrix} \boldsymbol{\Sigma}_X & \boldsymbol{\Sigma}_{XY} \\ \boldsymbol{\Sigma}_{YX} & \boldsymbol{\Sigma}_Y \end{pmatrix})$, where $\boldsymbol{\Sigma}_X = \boldsymbol{\Sigma}_Y = \boldsymbol{I}_{800}$, and $\boldsymbol{\Sigma}_{XY} = \rho \boldsymbol{\Sigma}_X (\boldsymbol{\phi} \boldsymbol{\eta}^T) \boldsymbol{\Sigma}_Y$. Here $\boldsymbol{\phi}, \boldsymbol{\eta} \in \mathbb{R}^{800}$, are sparse with 5 nonzero elements

| Linear model | | | |
|---|---|---|---|
| Method | $\hat{\rho}$ | $E_{\boldsymbol{\phi}}$ | $E_{\boldsymbol{\eta}}$ |
| PMA | 0.71 | 1.17 | 1.17 |
| SCCA | 0.90 | 0.060 | 0.066 |
| mod-SCCA | 0.90 | 0.056 | 0.062 |
| DG-CCA | 0.90 | **0.027** | **0.025** |

**Table 1:** Evaluating the estimation quality of the canonical vectors $\boldsymbol{\psi}$ and $\boldsymbol{\eta}$.

and $\rho = 0.9$. The indices of the active elements are chosen randomly with values equal to $1/\sqrt{5}$. In this setting, based on Proposition 1 in (Suo et al., 2017), the canonical vectors $\boldsymbol{a}$ and $\boldsymbol{b}$ which maximize the objective in Eq. 1 are $\boldsymbol{\phi}$ and $\boldsymbol{\eta}$ respectively.

Using this model we generate $400$ samples and estimate the canonical vectors based on CCA and DG-CCA. In Fig. 3 we present a regularization path of the proposed scheme. Specifically, we apply DG-CCA to the data described above using various values of $\lambda = \lambda_x = \lambda_y$. We present the $\ell_0$ of active gates (by expectation) along with the empirical correlation between the extracted representations $\widehat{\Sigma}_{XY}$, which is also $\hat{\rho} = \hat{\boldsymbol{\phi}} \boldsymbol{X} \boldsymbol{Y}^T \hat{\boldsymbol{\eta}}^T$. As evident from the left panel, there is a wide range of $\lambda$ values such that DG-CCA converges to true number of coefficients (10) and correct correlation value (0.9). Next, we present the values of $\boldsymbol{\phi}$, the DG-CCA estimate (using $\lambda = 30$) of the canonical vector $\hat{\boldsymbol{\phi}}$, and the CCA based estimate of the canonical vector $\hat{\boldsymbol{a}}$. The solution by CCA is wrong and not sparse, while the DG-CCA solution correctly identifies the support of $\boldsymbol{\phi}$. Finally, we evaluate the estimation error of $\boldsymbol{\phi}$ using $E_{\boldsymbol{\phi}} = 2(1 - |\boldsymbol{\phi}^T \hat{\boldsymbol{\phi}}|)$, and $E_{\boldsymbol{\eta}}$ is defined similarly. In Table 1 we present the estimated correlation along with the estimation errors of $\boldsymbol{\phi}$ and $\boldsymbol{\rho}$ (averaged over 100 simulations). As baselines we present the results simulated by Suo et al. (2017) (mod-SCCA), comparing the performance to PMA (Witten et al., 2009) and SCCA (Chen et al., 2013).

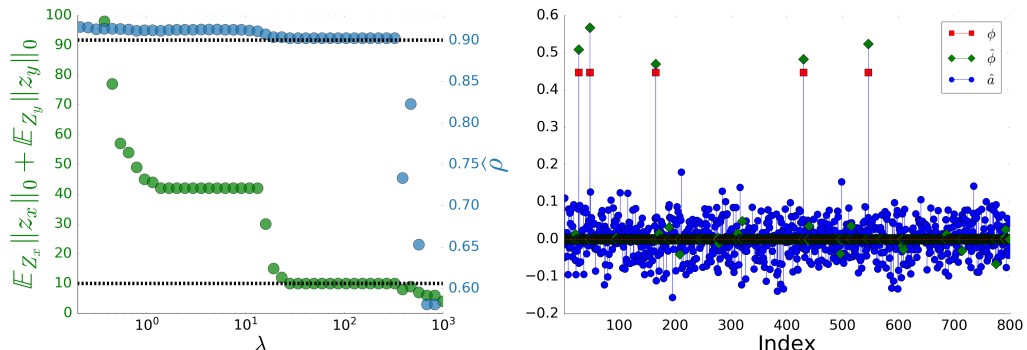

**Figure 3:** Left: Regularization path of DG-CCA on data generated from the linear model. The left $y$-axis (green) represents the sum of active gates (by expectation) after training. The right $y$-axis represents the empirical correlation between the estimated representations, i.e. $\hat{\rho} = \hat{\boldsymbol{\phi}}^T \boldsymbol{X}^T \boldsymbol{Y} \hat{\boldsymbol{\eta}}$, where $\hat{\boldsymbol{\phi}}$ and $\hat{\boldsymbol{\eta}}$ are the estimated canonical vectors. Dashed lines indicate the correct number of active coefficients (green) and true correlation $\rho$ (blue). Note that for small values of $\lambda = \lambda_x = \lambda_y$ the model select more variables than needed and attains a higher correlation value, this is a similar over-fitting phenomenon that CCA suffers from. Right: True canonical vector $\boldsymbol{\phi}$ along with the estimated vectors using DG-CCA ($\hat{\boldsymbol{\phi}}$) and CCA ($\hat{\boldsymbol{a}}$).

## 3.2 MULTI VIEW SPINNING PUPPETS

As an illustrative example we use a dataset collected by Lederman & Talmon (2018) for multiview learning. The authors have generated two videos capturing rotations of 3 desk puppets. One camera captures two puppets, while the other captures another two, where one puppet is shared across cameras. A snapshot from both cameras appears in the top row of Fig. 4. All puppets are placed on a spinning device that rotates the dolls at different frequencies. In both video there is a shared

parameter, namely the rotation of the common bulldog. Even thought the Bulldog is captured from a slightly different angle, we attempt to use CCA to identify a linear transformation that projects the two Bulldogs in to a common embedding. We use a subset of the spinning puppets dataset, with $400$ images from each camera. Each image has $240 \times 320 = 76800$ pixels (using a gray scaled version of the colored image), therefore, there are more feature than samples and direct application of CCA would fail. We apply the proposed scheme using $\lambda_y = \lambda_x = 50$, a linear activation and embedding dimension $d = 2$. DG-CCA converges to embedding with a correlation of $1.99$ using $372$ and $403$ pixels from views $\boldsymbol{X}$ and $\boldsymbol{Y}$. The active gates are presented in the bottom row of Fig. 4.

In Fig. 5 we present the coupled two dimensional embedding of both videos. Both embeddings are highly correlated with the angular orientation of the Bulldog. Note that adjacent images in the embedding are not necessarily adjacent in the original ambient space, this is because the Bunny and the Yoda puppets are gated and do not affect the embedding.

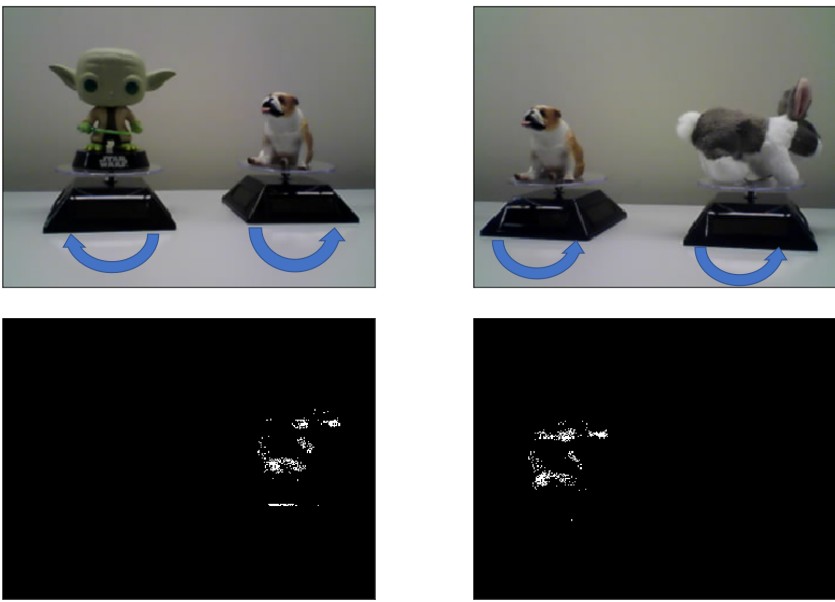

**Figure 4:** Top: two samples from the spinning puppets videos. Arrows indicate the spinning direction of each puppet. Bottom: the converged active gates for each video. There are $372$ and $403$ active gates for the left and right videos respectively.

### 3.3 NOISY MNIST

MNIST (LeCun et al., 2010) which consists of $28 \times 28$ grayscale digit images, with 60K/10K images for training/testing, is a well known and widely investigated dataset. We use a challenging variant of MNIST with coupled noisy views. The first view is created by adding noise drawn uniformly from $[0, 1]$ to all pixels. The second view is created by placing a random patch from a natural image in the background of the handwritten digits. Both views are based on different digit instances drawn from the same digit class. Random samples from both views are presented in Fig. 6. Both views consists of $62,000$ samples, of which we use $40,000$ for training $12,000$ for testing and $10,000$ are used as a validation set. Here the validation set is used for early stopping of the training procedure and optimizing $\lambda = \lambda_x = \lambda_y$.

Multi-view processing of the two noisy views can generate an informative representation of the noisy MNIST data. In the following we will focus on performing unsupervised embedding of each noisy MNIST view into correlated $10$ dimensional space.

By reducing the correlation cost, the DG-CCA learns which per-view pixels are relevant and informative in correlation maximization sense. In the bottom right corner of Fig 6 we present the location of the active gates. DG-CCA selects features (pixels) within an oval-like shape in the center of each view thus capturing the digit information and reducing the influence of the noise.

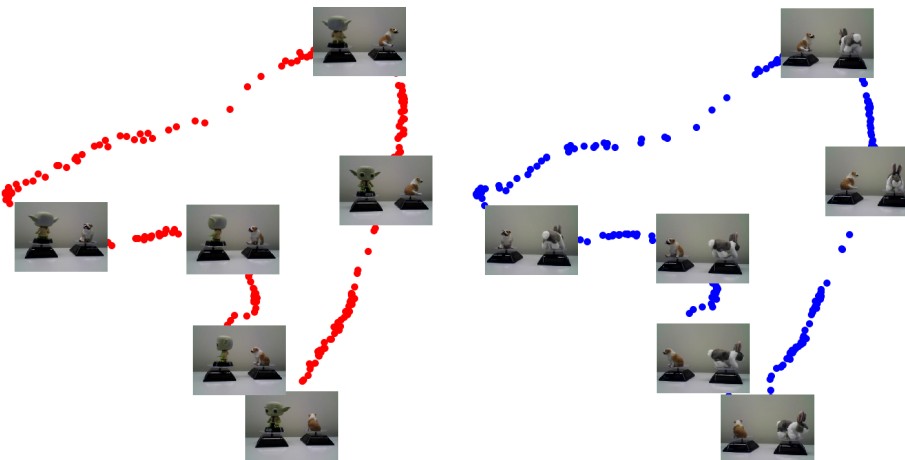

**Figure 5:** The two DG-CCA embedding of the Yoda+Bulldog video (left) and Bulldog+Bunny (right). We overlay each embedding with 5 images corresponding to 5 points in the embedding spaces. The resulting embeddings are correlated with the angular rotation of the Bulldog, which is the common rotating puppet in this experiment.

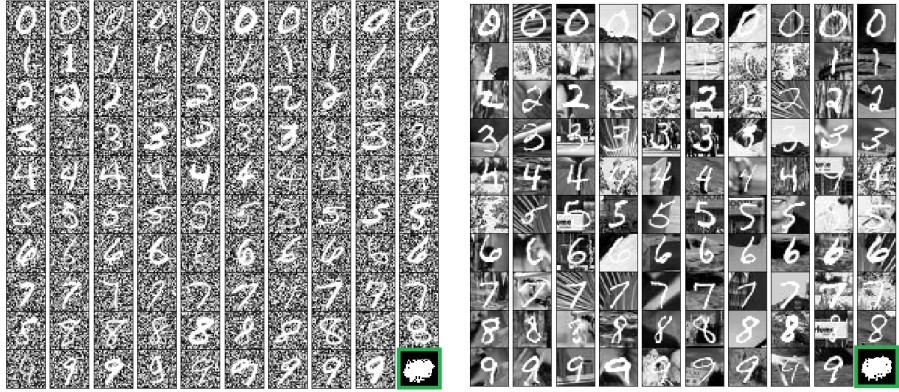

**Figure 6:** Images from noisy MNIST (left) and corresponding images from background MNIST (right). In the bottom right of both figures, we presents the active gates (white values within a green frame). There are 277 and 258 active gates for view I and II respectively.

To measure the class separation in the learned embedding, we apply $k$-means clustering to the stacked embedding of the two views. We run $k$-means (with $k = 10$) using 20 random initializations and record the run with the smallest sum of square distances from centroids. Given the cluster assignment, $k$-means clustering accuracy (KM ACC) and mutual information (MI) are measured using the true labels. Additionally, we train a Linear-SVM (LSVM) model on our train and validation datasets. LSVM classification accuracy (LSVM ACC) is measured on the remaining test set. The performance of DG-CCA compared with several baselines appears in Table 2. In the appendix we provide all implementation details and provide an experiment demonstrating the performance for various values of $\lambda = \lambda_x = \lambda_y$.

## 3.4 SEISMIC EVENT CLASSIFICATION

Next, we evaluate the method using a dataset studied by Lindenbaum et al. (2018). The data consists of 1609 seismic events. Here, we focus on 537 explosions which are categorized into 3 quarries. The events occurred between the years $2004 - 2015$, in the southern region of Israel and Jordan. Each event is recorded using two directional channels facing east (E) and north (N), these comprise the coupled views for the correlation analysis. Following the analysis by Lindenbaum et al. (2018), the input features are Sonogram representations of the seismic signal. Sonograms are time frequency

| Method | Noisy MNIST | | | Seismic | | |
|---|---|---|---|---|---|---|
| | MI | KM ACC (%) | LSVM (%) | MI | KM ACC (%) | LSVM (%) |
| Raw Data | 0.130 | 16.6 | 86.6 | 0.001 | 35.5 | 77.7 |
| CCA | 1.290 | 66.4 | 75.8 | 0.003 | 38.1 | 40.4 |
| SCCA | 0.342 | 23.9 | 63.1 | 0.610 | 71.7 | 86.9 |
| SCCA-HSIC | NA | NA | NA | 0.003 | 38.7 | 49.5 |
| KCCA | 0.943 | 50.2 | 85.3 | 0.006 | 38.4 | 92.5 |
| NCCA | 1.030 | 47.5 | 77.2 | 0.700 | 86.8 | 91.4 |
| DCCA | 1.970 | 93.2 | 93.2 | 0.830 | 94.9 | 94.6 |
| DG-CCA | **2.05** | **95.4** | **95.5** | **0.97** | **98.1** | **97.2** |

**Table 2:** Performance evaluation on the Noisy MNIST and seismic datasets.

representations with bins equally tempered on a logarithmic scale. Each Sonogram $z \in \mathbb{R}^{1157}$ with 89 time bins and 13 frequency bins. An example of Sonograms from both channels appears in the top row of Fig. 7.

We create the noisy seismic data by adding sonograms computed based on vehicle noise from [1]. Examples of noisy sonograms appear in the middle row of Fig 7. We omit 20% of the data as a validation set. Then we train DG-CCA to embed the data in 3 dimensions using several values for $\lambda = \lambda_x = \lambda_y$. Then, we use the model that attains maximal correlation on the validation set. In Table 2 we present the MI, $k$-means and SVM accuracies computed based on DG-CCA embedding. Furthermore, we compare the performance with several other baselines. Here, the proposed scheme improves performance in all 3 metrics while identifying a subset of 71 and 68 features from channel E and N respectively. The active gates are presented in the bottom row of Fig. 7.

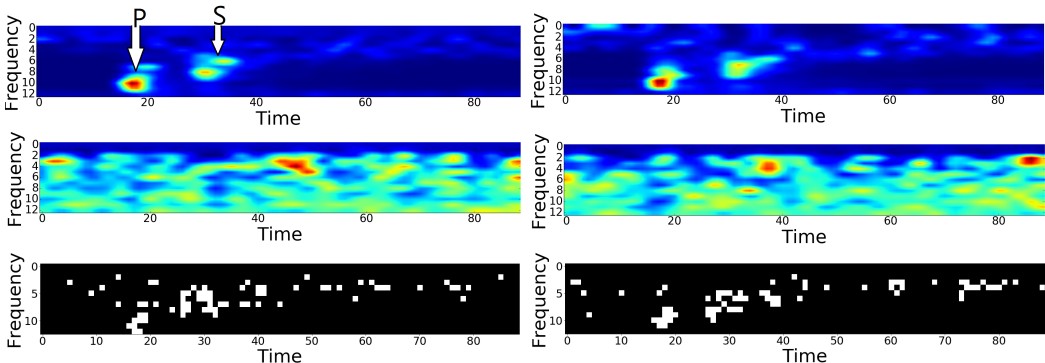

**Figure 7:** Top: Clean sample Sonograms of an explosion based on the E and N channels (left and right respectively). Arrows highlight the Primary (P) and Secondary (S) waves caused by the explosion. Middle: Noisy sonograms generated by adding sonograms of vehicle recordings. Bottom: the active gates for both channels. Note that the gates are active at time frequency bins which correspond to the P and S waves (see top left figure).

## 4 CONCLUSION

In this paper we present a method for learning sparse non-linear transformations which maximize the canonical correlations between two modalities. Our method is realized by gating the input layers of two neural networks which are trained to maximize their output's total correlations. Input variables are gated using a regularization term which encourages sparsity. This allows us to learn informative representations even when the number of variables far exceeds the number of samples. Finally, we demonstrate that the method outperforms existing methods for linear and non-linear canonical correlation analysis.

---

[1]https://bigsoundbank.com/search?q=car

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

## A APPENDIX

### A.1 GATES INITIALIZATION

The Gaussian based stochastic gates suggested by Yamada et al. (2020) are based on trainable parameters $\boldsymbol{\mu}$ and a constant parameter $\sigma$. These control the mean and standard deviation of the injected noise respectively. Yamada et al. (2020) have initialize all values of $\boldsymbol{\mu}$ to $0.5$, in this case, the gates approximate "fair" Bernoulli parameters. This is a reasonable choice, if no prior knowledge about the solution is available, however, we can utilize the closed form solution of the CCA problem to derive a smarter initialization procedure for the parameters of the gates. Specifically, given the empirical covariance matrix $\boldsymbol{C}_{XY} = \frac{\boldsymbol{XY^T}}{(N-1)}$, we denote the thresholded covariance matrix by $\boldsymbol{S}_{XY}$, with values defined as follows

$$(S_{XY})_{ij} = \begin{cases} (C_{XY})_{i,j}, & \text{if } |(C_{XY})_{i,j}| > \delta \\ 0, & \text{otherwise.} \end{cases}$$

Where $\delta$ is a threshold value selected based on the estimated sparsity of $\boldsymbol{X}$ and $\boldsymbol{Y}$. Specifically, if we assume that $r$ percent of the values should be zeroed, then $\delta$ is set to be the $r$-th percentile of $|(\boldsymbol{C}_{XY})|$. Then we compute the leading singular vectors $\boldsymbol{u}$ and $\boldsymbol{v}$ of $\boldsymbol{S}_{XY}$. We further threshold the absolute values of these vectors (using the same percentile used for $\boldsymbol{S}_{XY}$. The initial values of the parameters of the gates are then defined by $\boldsymbol{\mu}_X = \bar{\boldsymbol{u}} + 0.5$, and $\boldsymbol{\mu}_Y = \bar{\boldsymbol{v}} + 0.5$, where $\bar{\boldsymbol{u}}$ and $\bar{\boldsymbol{u}}$ are the thresholded versions of the absolute value of the singular vectors.

The standard deviation of the injected noise $\sigma$ was set to $0.5$ by (Yamada et al., 2020). They have selected this value as it maximized the gradient of the regularization term at initialization.

Empirically, we have observed that for DG-CCA smaller values of $\sigma$ translate to improved convergence. Specifically, we have used $\sigma = 0.25$ which worked well in our experiments. Studying the effect of $\sigma$ is an open question that we aim to pursue in future study.

## A.2 ADDITIONAL EXPERIMENTAL DETAILS

In the following sections we provide additional experimental details required for reproduction of the experiments provided in the main text.

### A.2.1 SYNTHETIC EXAMPLE

For the linear model we use a learning rate of $0.005$ with $10,000$ epochs. The values of $\lambda_x$ and $\lambda_y$ are both set to 30. These values were obtained using a cross validation procedure. We run the method 100 times with different realizations of $\boldsymbol{X}$ and $\boldsymbol{Y}$. Importantly, following Suo et al. (2017) we present the average errors for the estimation of the canonical vectors, however the median values are one order of magnitude better, specifically $E_{\boldsymbol{\psi}} = 0.0017$ and $E_{\boldsymbol{\eta}} = 0.0020$.

### A.2.2 NOISY MNIST

In this subsection we provide additional details regarding the noisy MNIST experiment. In Fig 8, we present the performance as a function of the number of active gates (pixels) controlled by $\lambda_x = \lambda_y$. The MI score, $k$-means and SVM accuracy were computed based on DG-CCA embedding with learning rate of $0.01$. Furthermore, the number of epochs ($\sim 4000$) was tuned by early stopping using random validation of size 10000. To learn 10 dimensional correlated embedding, we use the same architecture as suggested by (Wang et al., 2015) consisting of three hidden layers with 1000 neurons each. The number of dimensions in the embedding was selected based on the number of classes in MNIST. This architecture is used for both DCCA and DG-CCA. Note that for DG-CCA using small values of the regularization parameters $\lambda_x$ and $\lambda_y$, increases the number of selected features and the degrades performance. This is duo to the fact that as more features are selected more noise is introduced into the extracted representation (of size 10).It is interesting to note that the $k$-means was more robust to the introduced noise than the LSVM.

The regularization parameters $\lambda_x$ and $\lambda_y$ balances between the correlation loss and the amount of sparsification performed by the gates. These hyper parameters are tuned using the validation set in by maximizing the total correlation value. We compare DG-CCA to CCA (Chaudhuri et al., 2009)[2], KCCA (Bach & Jordan, 2002)[3], NCCA (Michaeli et al., 2016) [4] and DCCA (Andrew et al., 2013) [5]. For all methods we use an embedding with dimension 10, and evaluate performance with $k$-means using 20 random initilizations, and using LSVM by performing training on the training samples and testing on the remaining samples (split defined in the main text). In this experiment we tried to train SCCA-HSIC (Uurtio et al., 2018) [6] for over two days, but it did not converge. Furthermore, we believe that the poor performance of the kernel methods are degraded due to the high level of noise in the input images.

### A.2.3 SEISMIC EVENT CLASSIFICATION

Using the seismic data, we compare the performance of DG-CCA with a linear and non-linear activation. In this exaple, we use a learning rate of $0.01$ with 2000 epochs. The number of neurons for each hidden layer are: $300, 200, 100, 50, 40$, with a Tanh activation. he number of dimensions in the embedding ($d = 3$) was selected based on the number of classes in the data. Parameters are optimized manually to maximize the correlation on a validation set. In Fig. 9 we present SVM accuracy for different levels of sparsity. The presented number of features is the average over both modalities, and SVM performance is evaluated using 5-folds cross validation. We compare DG-CCA to CCA (Chaudhuri et al., 2009), SCCA (Suo et al., 2017), SCCA-HSIC (Uurtio et al., 2018), KCCA (Bach & Jordan, 2002) , NCCA (Michaeli et al., 2016) and DCCA (Andrew et al., 2013). For

---

[2]https://scikit-learn.org/stable/modules/generated/sklearn.cross_decomposition.CCA.html

[3]https://gist.github.com/yuyay/16ce4914683da30f87d0

[4]https://tomer.net.technion.ac.il/files/2017/08/NCCAcode_v3.zip

[5]https://github.com/adrianna1211/DeepCCA_tensorflow

[6]https://github.com/aalto-ics-kepaco/scca-hsic

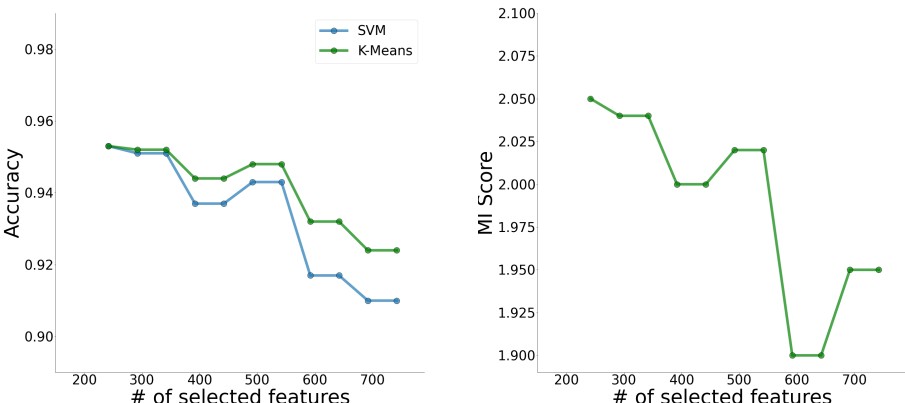

**Figure 8:** $k$-means and SVM classification accuracy (left) and mutual information score (right) vs. the number of selected features.

all methods we use an embedding with dimension 3, and evaluate performance with $k$-means using 20 random initilizations, and using linear SVM by performing a 5-folds cross validation. For the kernel methods we evaluated performance by constructing a kernel using $k = 5, 10, ..., 50$, nearest neighbors and selected the value which maximized performance in terms of total correlation.

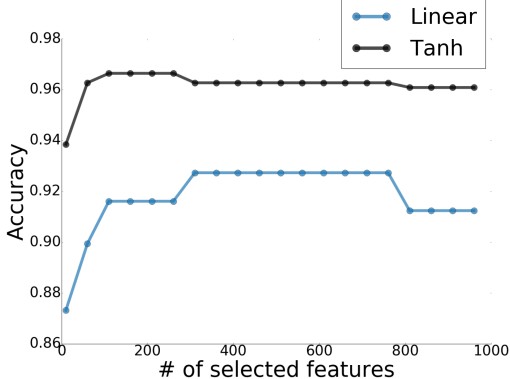

**Figure 9:** Classification accuracy on the noisy seismic data. Performance is evaluated using linear SVM in the 3 dimensional embedding. Comparing performance of DG-CCA for different levels of sparsity, and using linear and nonlinear activation (Tanh).

