# OpenReview forum: "Deep Gated Canonical Correlation Analysis"
_ICLR.cc/2021/Conference — Reject_

### Official Review · AnonReviewer1 · 2020-10-28
**Sparse Deep CCA variant building on existing components with good but somewhat partial empirical evaluation**

**Rating:** 4
**Confidence:** 4

**Review:**

Summary: The authors propose a new non-linear CCA variant that learns mappings that are sparse with respect to the input variables, using approximate $l_0$ regularisation, to improve performance for applications with large number of features but few samples.

Reasons for score: I am leaning towards rejection due to the straightforward nature of the work. The method combines two existing techniques in fairly obvious way and despite good empirical comparisons has also issues in evaluation since more recent comparison methods are missing.

Detailed feedback: The related work and importance of the application are well covered, and the technical solution is sound. The conceptual novelty is, however, fairly limited; several sparse CCA variants have been proposed in the past and switching to proper sparsity ($l_0$ vs more common $l_1$) is a natural thing to do. Furthermore, in recent years technical solutions building on stronger sparsity priors have been proposed for closely related models (e.g. Boyveyron et al. "Bayesian variable selection of globally sparse probabilistic PCA" (2018) discusses this in detail for PCA, and many algorithmic details for PCA generalise easily for CCA by interpreting CCA as group-sparse PCA).

The specific technical solution presented here appears to be new, but builds directly on existing and relatively obvious choices: The loss matches Andrew et al. (2013) and the $l_0$ approximation is from Yamada et al. (2020). Even though the specific formulation in the latter is recent, the underlying auxiliary variable construct has been used for similar purposes before. The simplicity of the technical approach is highlighted by the fact that the whole model description takes only slightly more than one page of the paper. In summary, the paper does not make fundamental conceptual or technical contributions. It certainly has potential for being a useful practical tool for the task, but the required scientific insight is limited.

The empirical demonstrations are nice and illustrative, but carried out on somewhat simplified benchmark data. They do show that the method works well in comparison against reasonably chosen competing methods, but do not clearly indicate qualitative change in CCA applications. The advantage over ordinary DCCA, published already 7 years ago, is not particularly striking in Table 2, and in recent years quite a few deep CCA variants have been proposed but are not compared against (or cited). Consequently, we cannot really evaluate whether this advances the field in practice; there is potential, but as it is the empirical comparisons do not seem sufficient to overcome the lack of technical and conceptual contribution.

---

### Official Review · AnonReviewer2 · 2020-10-29
**Gated CCA**

**Rating:** 4
**Confidence:** 3

**Review:**

This paper combines an approximate $L_0$ regularization on the canonical vectors with CCA to encourage the CCA for getting sparse vectors. In addition, the CCA is computed on embeddings from a neural network, which make it possible to capture non-linear correlations.

Overall, the paper is well written and easy to follow. The paper seems to be a combination of deep CCA (Andrew et al. 2013) and Louizos et al. 2017. In particular, the $L_0$ regularization approximation is very similar to that proposed in Louizos et al. 2017. It would be great if the authors could be more clear on illustrating the differences (if any). Therefore, the novelty of this paper is unclear.

The experiments could be improved. Since most of the experiments were carried out on relatively small datasets with reasonable sized model, it would be great to have multiple runs that illustrate the stability/variance of the method. In addition, the major benefit of using neural networks as embedding function is the ability to capture non-linear relationships. It would be great to add a synthetic example to illustrate this benefit. The authors mentioned the use of early stopping and hyper-parameter selection, however, it is not clear based what criteria those actions were carried out. My guess is that it is based on the objective in eq. 4 on the validation set. It would be great if the authors could make this clear, because from the synthetic experiments, $\lambda$ plays a quite important role for the final performance.

---

### Official Review · AnonReviewer4 · 2020-11-01
**The paper presents a new deep CCA method that applies gating to input variables using a latent clipped Gaussian random variable to avoid overfitting. The total novelty and technical contributions seems limited.**

**Rating:** 5
**Confidence:** 3

**Review:**

This paper presents a new deep CCA method to learn non-linear relationships between two modalities. It trains two neural networks each for a modality to maximize the total correlations of their output representations. Gating is applied to input variables by associating each with a latent Bernoulli variables which is then relaxed with the clipped Gaussian random variable. Experiments on one synthetic and two real datasets demonstrate the superiority of the proposed method.

Below are specific comments.

1. In the last sentence of Section 2.2, it is unclear to me how the stochastic part of the gates is removed to determine whether $z_x[i_x]$ is equal to or larger than zero. Is $z_x[i_x]$ determined based on the estimated $\mu_x[i_x]$: $z_x[i_x] > 0$ if $\mu_x[i_x] > 0$ and $z_x[i_x] = 0$ otherwise?

2. Minor comments (notation inconsistencies/abuse, typos, etc.):

The sentence "For example, in biology ... and engineering (Chen et al., 2017)" is not complete (sentence fragment). Please rephrase it or join it to the preceding sentence.
Is "the degeneracy inherit to $N < D_x,D_y$" supposed to be "the degeneracy inherent to $N < D_x,D_y$"?
The word "interpetability" is misspelled.
In the middle subfigure of Figure 1, it is clearer if the label $\epsilon$ and tick values $\{-0.5,0,0.5\}$ are added along the horizontal axis.
"straight forward" should be spelled as "straightforward" (no space).

Eq. (4): It seems $\boldsymbol{z}_x^T \boldsymbol{X}, \boldsymbol{z}_y^T \boldsymbol{Y}$ should be written as $\mathop{{\rm diag}}\left(\boldsymbol{z}_x\right) \boldsymbol{X}, \mathop{{\rm diag}}\left(\boldsymbol{z}_y\right) \boldsymbol{Y}$. Note that the $\boldsymbol{X}, \boldsymbol{Y}$ here represent the observed data matrices of dimensions $D_x \times N, D_y \times N$ [rather than random vectors based on which the data are observed].

In Section 2.2, third line, the expression of the regularization:
- $\mathbb{P}(\boldsymbol{z}_x[i] \geq 0)$ should be $\mathbb{P}(\boldsymbol{z}_x[i] > 0)$ or $\mathbb{P}(0 < \boldsymbol{z}_x[i] \leq 1)$;
- For consistency, it should write $\|\boldsymbol{z}\|_0$ as $\|\boldsymbol{z}_x\|_0$ and the index $i$ as $i_x$.

"a similar notations" should be "similar notations".

In Section 2.2, first paragraph, last sentence "The total correlation in Eq. 4 can be expressed using the trace of ..."
- "total correlation" should be "total squared correlation".

In Section 3.1, second paragraph, $\hat{\rho}=\bm{\hat{\phi}}\boldsymbol{X}\boldsymbol{Y}^T\bm{\hat{\eta}}^T$ should be $\hat{\rho}=\bm{\hat{\phi}}^T\boldsymbol{X}\boldsymbol{Y}^T\bm{\hat{\eta}}$.

---

### Official Review · AnonReviewer3 · 2020-11-01
**A method for deep sparse canonical correlation analysis**

**Rating:** 5
**Confidence:** 4

**Review:**

1. Paper summary:

This paper proposes a DL method for learning sparse non-linear transformations that maximize correlations between two views. In particular, each view is passed through a separate network. Stochastic Gating is applied to the input layer of each network. The two networks are jointly trained by maximising the correlation between their outputs. Sparsity is obtained by imposing L0 regularization terms on the Stochastic Gating variables.

2. Strong points of the paper:

Stochastic Gating gives way to an objective function that can be optimized through Stochastic Gradient Descent.

The method can detect correlation between two views even when data size is less than the number of dimensions, as demonstrated by the experimental results.

3. Weak points of the paper:

The proposed method is very similar to DCCA paper of Andrew et al. The only difference is Andrew et al. use L2 regularization, while the authors use L0 regularization.

Similarly to DCCA, the method suffers from the two issues.

First, the method learns non-linear transformations that however are hard to interpret. Non-linear CCA can be achieved by learning linear transformations through non-linear correlation measure, such as HSIC. HSIC-CCA [1] can also learn sparse representations. Given the rising importance of explainable AI, non-linear transformations seem to be a drawback.

Second, the method relies on Stochastic Gradient Descent. However, the loss function is not decomposable into batches. This makes batch training somewhat random.

4. Conclusion:

For the above reasons, I find the contributions of the paper to be marginal.

[1] Billy Chang, Uwe Krüger, Rafal Kustra, Junping Zhang: Canonical Correlation Analysis based on Hilbert-Schmidt Independence Criterion and Centered Kernel Target Alignment. ICML (2) 2013: 316-324

---

### Decision · Program_Chairs · 2021-01-07
**Final Decision**

**Decision:**

Reject

**Comment:**

The paper proposes an approach to sparse CCA with deep neural nets, performing simultaneous feature selection with stochastic gating and canonical correlation maximization.  The reviewers think that there is merit in defining an objective function that optimizes the goals jointly throughout the networks. However, the paper has not clearly presented the novelty in methodology. In particular, the reviewers agree that the paper needs to clearly distinguish itself from the two building blocks (Andrew et al. 2013 and Louizos et al. 2017), and demonstrate the significance of combining the two techniques theoretically and/or experimentally. Also, there is a large literature in sparsifying classical method. Sufficient discussions and comparisons with prior work can better position the current work in the literature.